# Experimental Investigation on the Influence of Swirl Ratio on Tornado-like Flow Fields by Varying Updraft Radius and Inflow Angle

Pengfei Lv [1], Yumeng Zhang [1,2,*], Yanlei Wang [1] and Bo Wang [1,*]

1 Key Laboratory of Western China's Environmental Systems (Ministry of Education) and Engineering Research Center of Fine Particle Pollution Control Technology and Equipment, College of Earth and Environmental Sciences, Lanzhou University, Lanzhou 730000, China

2 College of Atmospheric Sciences, Lanzhou University, Lanzhou 730000, China

* Correspondence: zhangyumeng@lzu.edu.cn (Y.Z.); wangbo@lzu.edu.cn (B.W.); Tel.: +86-188-9379-8184 (Y.Z.); +86-151-0131-1520 (B.W.)

**Abstract:** The swirl ratio is the most critical parameter for determining the intensity and structure of tornado-like vortex, defined as the ratio of angular momentum to radial momentum. The angle of entry flow and the updraft radius are two key parameters affecting the swirl ratio. Many laboratory simulators have studied the effect of swirl ratio by changing the angle of entry flow, but there is a lack of research on the updraft radius. Therefore, for a deep sight of the impact of the updraft radius on the swirl ratio and tornado-like vortex, a laboratory tornado simulator capable of adjusting the updraft radius was designed, built, and tested. And, the effects of various swirl ratios caused by the updraft radius and the angle of entry flow on the tornado-like vortices were investigated, in terms of the dual-celled vortex transformation and vortex wandering. It was found that the effects of the updraft radius and the angle of turning vanes on the tornado-like vortices are quite different, and the formation of the dual-celled vortex is more sensitive to the updraft radius, because a larger angular momentum and axial pressure gradient can be provided. In addition, increasing the updraft radius has a greater inhibitory effect on the vortex wandering phenomenon compared to the angle of the turning vanes due to the flow fluctuations induced by turbulence.

**Keywords:** tornado-like vortices; swirl ratio; updraft radius; laboratory simulations; particle image velocimetry

## 1. Introduction

Tornadoes are among the most devastating natural hazards with high wind speeds that cause severe damage to structures [1,2]. According to statistical records, the number of tornadoes per tornado event in the United States exhibited a differential growth at a rate of 2.89% per year from 1954 to 2014 [3]. Additionally, the incidence of tornadoes has increased in countries that are not traditionally prone to tornadoes, such as Japan and China, due to global climate change [4]. This has led to a series of scientific studies on tornadoes. The dynamical research was predominantly focused on the conditions of tornado generation, fluid structure and damage properties [5,6]. The current research methods mainly include field observations, numerical modeling and physical simulation using a vortex chamber or tornado simulator in the laboratory.

Due to the unpredictable nature and extreme destructiveness of tornadoes, direct observations of near-ground tornado flow fields pose considerable challenges. The advancement of mobile Doppler radar technology has enabled the safe monitoring and investigation of tornadoes in their natural environment. However, radar waves do not follow the curvature of the Earth, and due to obstructions from ground-based objects, Doppler radar cannot directly measure areas very close to the ground [7]. Bluestein et al. [8,9] utilized

onboard mobile Doppler radar to achieve long-range tracking and monitoring of tornadoes, recording both tangential and radial velocities of multiple tornadoes. They also conducted preliminary statistical analyses of wind speeds at different altitudes. Similarly, Alexander [10] and Wurman [11] employed Doppler radar to observe tornadoes that occurred in the Spencer region in May 1998. They obtained velocity distribution information at different heights and times, which provided field-measured data for subsequent numerical and laboratory simulations. Hocker [12] suggested the presence of downdrafts at the centers of tornado vortices and deduced tangential and radial velocity fields in certain regions within the vortex. Similarly, Markowski emphasized the significant role of downdrafts in tornado dynamics. In the early stages of tornado formation, if the vertical vorticity near the ground is weak, a near-ground downdraft is required to enhance vertical vorticity stretching and circulation. Once a tornado stabilizes, the downdrafts influence the tornado's morphology and circulation distribution.

With the emergence and advancement of computers, numerical simulation techniques have become increasingly integral to the exploration of tornado vortices. In 1975, Wen [13] considered the velocity distribution within the boundary layer and formulated a mathematical model to depict tornado phenomena. This model was subsequently juxtaposed with actual observational data from the 1957 Dallas tornado, resulting in a noteworthy congruence between the two. Building upon an axisymmetric framework, Rotunno [14] simulated the internal flow structures within a Ward-type simulator, delving into the vortex dynamics of tornado-like systems. The findings underscored the pivotal role of the swirl ratio (S) as a vital parameter impacting vortex architectures within the simulator. Lewellen [15] employed Large Eddy Simulation (LES) to probe the interactions between single-vortex tornadoes and the ground. Their investigation unveiled the formation of secondary vortices proximate to the vortex core, providing a basis for elucidating the mechanisms underpinning tornado rotation and motion. Hangan [16] grounded in the Reynolds-averaged Navier–Stokes (RANS) equations, simulated tornado vortices, discerning the influences of the swirl ratio on tornado vortex dynamics, with an intent to establish a correlation between the swirl ratio and the Fujita scale. Yuan [17] employed modeling of the ISU-type tornado simulator, yielding static and translating tornado flow fields for comprehensive characterization of overall and near-ground flow features. Moreover, Verma [18] investigated various open-system Ward-type tornado simulators, discovering an increase in internal pressure with gradual restriction of the outlet dimensions. In 2023, Zhao [19] examined the effects of a series of swirl ratios and radial Reynolds number gradients on multi-vortex tornadoes. The outcomes indicated that augmenting the swirl ratio led to an increase in sub-vortex count, enlargement of dimensions, and reduction of maximum tangential velocity. Additionally, while increasing the radial Reynolds number did not affect sub-vortex count, it diminished sub-vortex dimensions while increasing rotational speed.

Tornado-like flows were first experimentally simulated by Ying and Chang in 1970 [20]. They employed a rotating cylindrical screen to generate circulation above the ground, while an updraft was induced using a suction fan. Subsequently, Ward [21] enhanced the design in 1972 by introducing the Ward-type Tornado Vortex Chamber (TVC). In this design, convergent airflow resulted from the rotation of the screen at the chamber's perimeter, while a convective flow was generated through an adjustable-speed fan. Moreover, Church [22] achieved a more stable vortex and a sequence of tornado vortex configurations by increasing the angular momentum input to the swirling flow, leading to the development of the Purdue larger Ward-type laboratory tornado vortex simulator. Another innovation is a moveable tornado simulator (Iowa State University, ISU) built by Hannin in 2008 [23]. This apparatus featured an axial downward velocity in the inflow region, inspired by observations of rear-flank downdrafts (RFDs) surrounding regions of intensified low-level vorticity in actual tornadoes, potentially playing a significant role in near-surface tornado generation [24]. Furthermore, the Wind Engineering Energy and Environment (WindEEE) Dome, constructed by the University of Western Ontario in 2016, presents a unique, large-scale hexahedral wind tunnel. This facility has the capability to generate

various wind environments, including translatable tornadoes, downbursts, and various boundary-layer winds [25]. Research on these laboratory simulators has greatly improved the understanding of tornado flow fields.

In these studies, radial Reynolds number and swirl ratio were recognized as two dimensionless groups to control the tornado structure. Specifically, radial Reynolds number characterizes the relative influence of inertial and viscous forces, while the swirl ratio quantifies the rotational intensity of the flow [22,26]. In addition, the aspect ratio is an important geometric dimensionless parameter that is commonly used to evaluate the size ratio of tornado simulators. Understanding these parameters is essential as they significantly impact various aspects of tornado behavior, including the performance of tangential velocity, vortex core structure, and pressure drop. Regardless of the simulator type, the way most laboratory simulations change the radial Reynolds number and swirl ratio was similar. Table 1 summarizes some important simulators used in recent years to study the tornado-like vortices, including their characteristics and variable parameters. In general, the radial Reynolds number was changed by varying the fan speed and thus the updraft flow rate, and the swirl ratio was changed by varying the angle of the turning vanes and thus the angular momentum.

**Table 1.** Summary of the characteristics and structure of the three tornado simulators.

| Expermental Tornado Simulator | Author and Institution | Characteristics | Variable Parameters |
|---|---|---|---|
| ISU simulator [23] | Iowa State University, Haan et al. | • The turning vanes were placed at the top of the tornado generator, and it employed a "rotating forced downdraft" to loosely follow the real-life tornado observations.<br>• The suspended simulator was able to be moved. | • Vane angles (0–55°)<br>• Floor heights, (i.e., different distances from the ground plane to the downburst duct)<br>• Fan speed |
| VorTECH simulator [27] | Texas Tech University, Tang et al. | • "Ward Type" design.<br>• Large-scale, it has a chamber of 10.2 m in diameter, an updraft hole of octagonal cross-section 4 m in diameter. | • The orientations of the turning vanes<br>• The speed of the fans<br>• The heights of the chamber and the turning vanes (1–2 m) |
| Wind Engineering Energy and Environment (WindEEE) Dome [25] | Western University Refan et al. | • The simulator has over 100 independently controllable fans which allows it to produce a variety of flows.<br>• Large, three-dimensional and time-dependent wind testing chamber, outer diameter can reach 40 m.<br>• The tornado can be translated at a maximum speed of 2 m/s over a distance of 5 m through the chamber. | • The angle of vanes at the periphery<br>• Top fans speed |

Based on the laboratory simulator, it was extensively shown that for a given geometry and for a smooth surface, above a certain critical value of radial Reynolds number, the core radius and the transition from a single vortex to multiple vortices are independent of the radial Reynolds number and are strongly a function of the swirl ratio. Thus, the swirl ratio has received a lot of attention because it is regarded as the most important parameter to control the tornado structure. There are several theoretical definitions of swirl ratios, in terms of structural parameters, including the angle of entry flow and the radius of the

vortex core [28]. However, limited by the structure of large equipment, only the swirl ratio variation through the angle of entry flow was investigated, and the different swirl ratios caused by vortex radius and its effect have not been discussed in depth.

Therefore, this study designed a small-scale tornado simulator, which was able to change the radius of the updraft in addition to changing the parameters of traditional equipment. Further, the properties of the tornado-like flow field, including tangential velocity, core radius, vortex transformation and vortex wandering, were investigated by the Particle Image Velocimetry (PIV) for different updraft radii and angles of entry flow. This study can provide more insight into the effect of the swirl ratio on the tornado flow field.

The remainder of the paper is organized as follows. The tornado simulator and the PIV system are briefly described in Section 2. The validation of the simulator is in Section 3. Sections 4 and 5, respectively, describe two methods to change the swirl ratio and its effect on the tornado-like flow. Their results were then compared and the main findings are finally summarized in the conclusion, Section 6.

## 2. Simulator and Experiment Setup

### 2.1. Simulator Design

The simulator was based on a simplified approach to the atmospheric dynamics assumptions for tornado generation, which ensured the minimum necessary conditions for tornado generation, namely (1) the air flow in the vertical direction at ground level and (2) the wind shear of air flow in the horizontal direction [29,30]. The dimensions of the simulator were based on the six relatively independent dimensionless numbers proposed by Davies-Jones according to the Buckingham $\pi$ Theorem [31]; the schematic illustration and dimensions of the simulator are shown in Figure 1. To visualize the flow field, the simulator structure was designed as a hexagonal column with transparent and high-strength Plexiglas material. Optical glass with better light transmission was used on the walls where the laser passes through, which improved the accuracy of optical measurement instruments such as PIV.

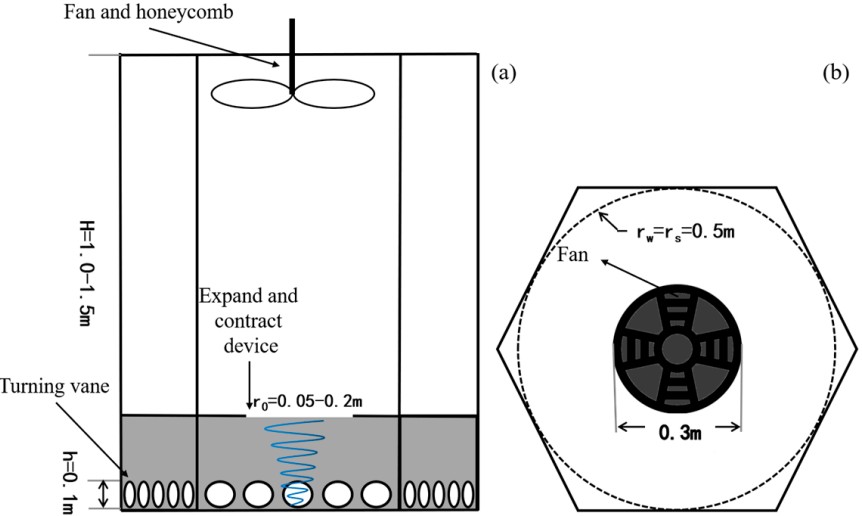

**Figure 1.** Diagram of the tornado simulator: (**a**) front view; (**b**) top view.

This simulator has a chamber 0.5 m in radius, an updraft hole with a variable diameter hexagonal cross section, 30 turning vanes and independent small fans in the periphery of the chamber, a large fan at the top, and a honeycomb that functions as the baffle. To generate flows of a desired structure, the orientations of the turning vanes can be varied between 0° and 60° to control the angular momentum of the inflow, the speed of the fans can be varied to control the amount of updraft, and the heights of the chamber can be adjusted between 0.6 and 1.1 m to control the internal aspect ratio of the apparatus. It was worth noting that the upper part of the inlet area was equipped with an expand-and-contract device to

change the radius of the updraft. The design of the device was based on the diaphragm structure and the radius variation range was from 50 mm to 200 mm. In addition, all these variable parameters were integrated into a self-made software by means of electric devices and Bluetooth transmission, which allowed real-time control of the simulator parameters through the cell phone.

### 2.2. Experiment Setup

The experimental platform in this study is shown in Figure 2a, and the measurement techniques mainly include particle image velocimetry (PIV) and high-speed camera. The high-speed camera used in this experiment was the Revealer 5F01, which was based on the CMOS image sensor and can achieve a frame rate of 2000 fps at full-frame resolution (1280 × 1024 dpi). And, the two-dimensional PIV measurement system produced by TSI was employed to provide full flow-field-velocity information. Moreover, the glycol solution was atomized by an ultrasonic atomizer to produce droplets of uniform concentration and size to provide tracer particles for PIV systems. These particles were sufficiently small to follow the fluid motion accurately and not alter the fluid properties or flow characteristics. Figure 2b demonstrates the PIV system components as well as the test setup. The horizontal velocity field (radial and tangential components) were measured at the center of the simulator and at 25 cm heights above the ground. The red arrows indicate the camera's viewing perspective.

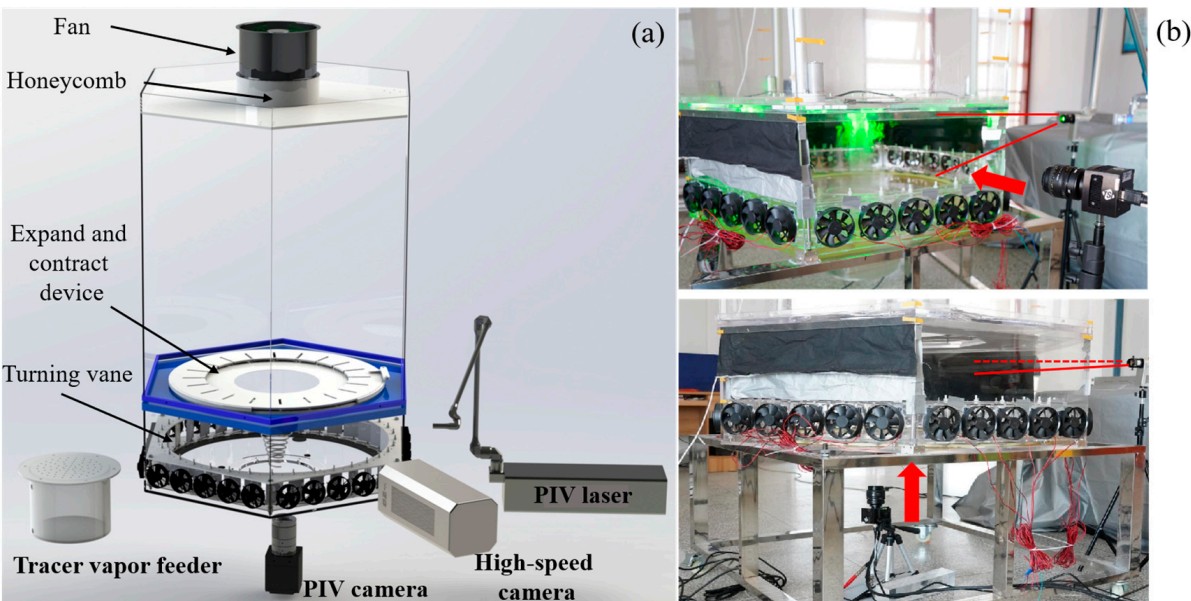

**Figure 2.** Experiment platform: (**a**) schematic diagram; (**b**) PIV setup.

## 3. Controlling Parameters and Validation

### 3.1. Controlling Parameters

As is now well recognized, three important flow parameters control the dynamics and geometry of simulated tornado-like vortices including the radial Reynolds number, the swirl ratio and the aspect ratio [32]. The radial Reynolds number is defined as

$$Re_r = \frac{Q}{2\pi\nu} \tag{1}$$

where $Q$ is a volume flow rate per unit axial length and $\nu$ is the kinematic viscosity of the fluid. The radial Reynolds number of a real tornado is about $10^9$ to $10^{11}$, and laboratory simulated tornado-like vortices have difficulty reaching this level. However, it was previously shown that when the Reynolds number is large enough (generally

$4.1 \times 10^3 \sim 1.2 \times 10^5$ in tornado simulators), the core radius and the transition from a single vortex to multiple vortices are independent of $Re_r$ and are strongly a function of swirl ratio [25]. The range of radial Reynolds number variation in this simulator was designed to be $3.5 \times 10^4$–$6.5 \times 10^4$.

Aspect ratio is defined as the ratio of inflow height to radius of the updraft hole, as in Equation (2) below.

$$a = \frac{h}{r_0} \tag{2}$$

where $h$ and $r_0$ are the depth and radius of the convergence region of the flow, which correspond to the height of the turning vanes and the radius of the updraft hole of the facility, respectively.

The swirl ratio is basically defined as the ratio of angular momentum to radial momentum in the vortex, and is expressed in Equation (3).

$$S = \frac{r_0 \Gamma_\infty}{2Qh} = \frac{\Gamma_\infty}{2aQ} \tag{3}$$

where $\Gamma_\infty$ is the free-stream circulation at the outer edge of the convergence region. Furthermore, for the Ward-type tornado simulator with the turning vanes placed at the bottom. The original swirl ratio $S$ can be simplified by replacing the ratio of free-stream circulation to volume flow rate with $\tan \theta$, where $\theta$ is the angle of the turning vanes.

$$S = \frac{\tan \theta}{2a} \tag{4}$$

In addition, the aspect ratio also varies with the radius of the updraft, but according to previous experimental investigations, this effect can be neglected [31,32].

### 3.2. Validation Objectives

The primary objective of the validation work was to ensure that the velocity field generated by the tornado simulator matched that of an actual tornado. Figure 3 displays flow images captured by a high-speed camera at different swirl ratios and time instants, with a temporal resolution of 13 ms. Panels (a) and (b) in Figure 3 correspond to swirl ratios of 0.13 and 0.43 for t = 1ms, t = 14 ms, t = 27 ms, and t = 40 ms, from left to right. The tornado-like vortex structure exhibited an overall columnar shape extending from the ground to the updraft orifice, with minimal variation in vortex core size with height observed in both cases. At a swirl ratio S = 0.43 (Figure 3b), the vortex diameter was larger, and the distortion of the vortex core was more pronounced. As labeled by the arrows in the figure, the vortex exhibited evident bending and deformation. These observations were attributed to the adverse axial pressure gradient [33] and the wandering effect [34]. Overall, the captured flow patterns demonstrated remarkable agreement with previously simulated vortices in tornado vortex chambers [21,25] and real tornadoes [30].

In addition, the normalized tangential velocity along the radial direction was compared with the Spencer and Mulhall tornados [10,11], as well as Rankine and Oseen vortex theoretical models [35,36]. As can be seen from Figure 4, the tangential velocities at different swirl ratios exhibited a typical pattern of increasing and then decreasing from the center outward, which was in good agreement with the measured data and the theoretical model, demonstrating the reliability of this simulator. In addition, the data for the Rankine vortex were slightly inconsistent because they did not consider viscous effects such as vortex diffusion and energy dissipation.

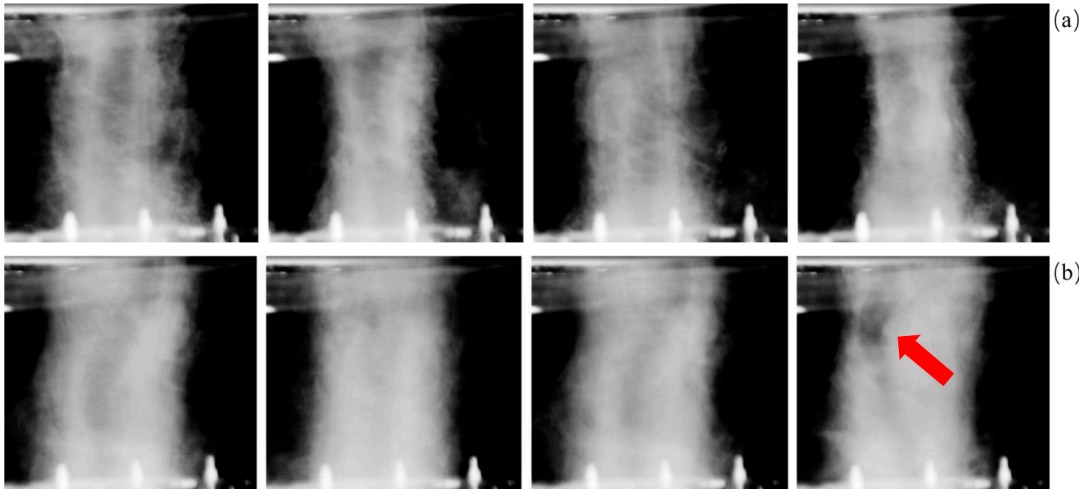

**Figure 3.** Tornado-like vortex produced in simulator: (**a**) *S* = 0.13; (**b**) *S* = 0.43 with a temporal resolution of 13 ms. Panels from left to right are t = 1ms, t = 14 ms, t = 27 ms, and t = 40 ms.

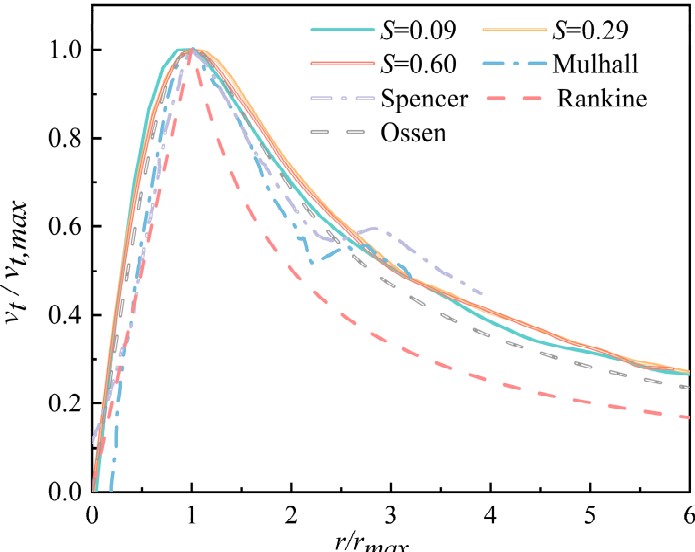

**Figure 4.** Normalized tangential velocities vs. radius.

## 4. Different Swirl Ratios Caused by the Angle of the Turning Vanes

After validation, we took two approaches to change the swirl ratio and investigated its effect on the tornado-like flow, starting with the angle of the turning vanes. Three cases were studied and they were divided into three groups according to the fixed updraft radius, as shown in Table 2.

**Table 2.** Group sets with different angle of the turning vanes.

| Group | Updraft Radius $r_0$ (cm) | Angle of the Turning Vanes $\theta$ (°) | Swirl Ratio $S$ |
|---|---|---|---|
| 1 | 10 | 10, 20, 30, 40, 50, 60 | 0.09, 0.18, 0.29, 0.42, 0.60, 0.87 |
| 2 | 15 | 10, 20, 30, 40, 50, 60 | 0.13, 0.27, 0.43, 0.63, 0.89, 1.30 |
| 3 | 20 | 10, 20, 30, 40, 50, 60 | 0.18, 0.36, 0.58, 0.84, 1.19, 1.73 |

Figure 5 displays the instantaneous streamlines superimposed on the horizontal velocity contours of group 1. As shown, the vortices observed at different swirl ratios are all counterclockwise rotating single-celled vortexes. With $\theta$ from 10° (a) to 60° (d), vortex

radius and velocity are increased; this is because *S* can stabilize the vortex structure in this range. However, as $\theta$ continues to increase, the vortex radius increases, and the velocity decreases. The average tangential velocity and vortex radius are shown in Figure 6 to illustrate the phenomenon more clearly. The reason is the effect of vortex breakdown, high swirling velocities and pressure deficit which create a vertical pressure gradient that causes downward flow, further causing the vortex radius to increase and velocity to decrease. At this time, the vortex is in a critical state and no dual-celled vortex has been observed.

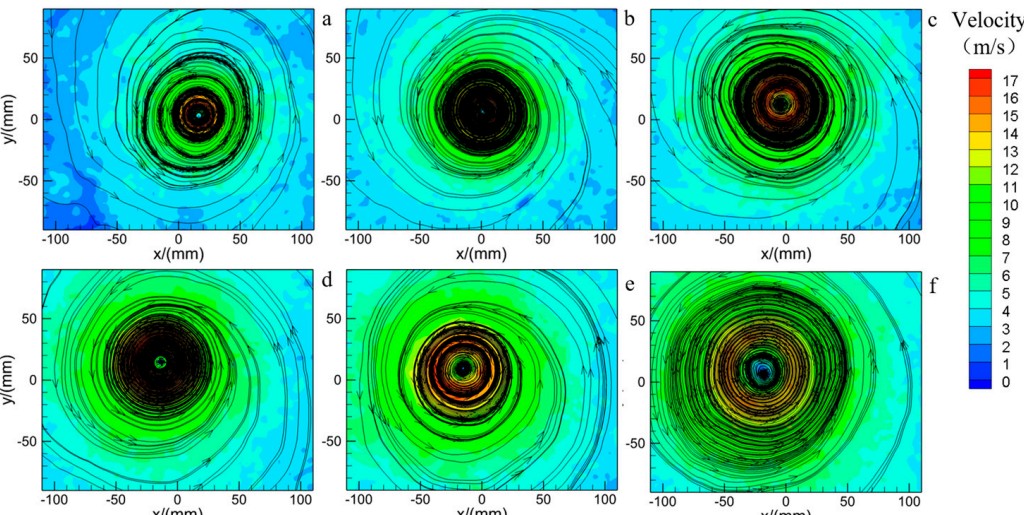

**Figure 5.** Instantaneous streamlines superimposed on the horizontal velocity contours of group 1: (**a**) $\theta = 10°$; (**b**) $\theta = 20°$; (**c**) $\theta = 30°$; (**d**) $\theta = 40°$; (**e**) $\theta = 50°$; (**f**) $\theta = 60°$.

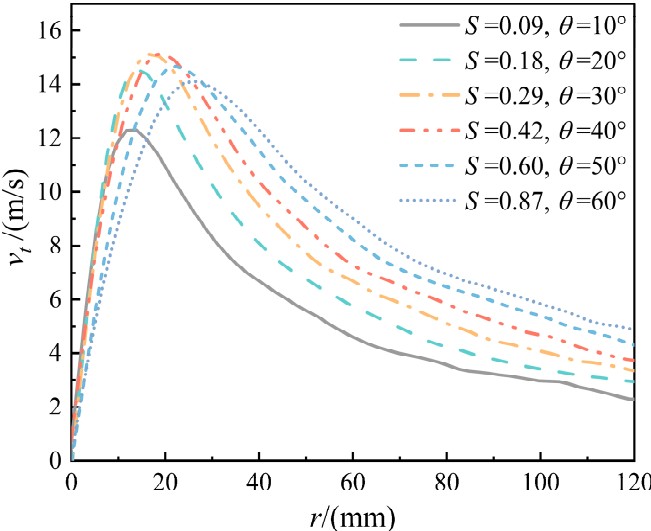

**Figure 6.** Mean tangential velocity along radial for different swirl ratios.

Figures 7 and 8 show the results of group 2 and group 3. As can be observed in Figure 7 ($r_0 =5$ cm), *S* reaches 0.43 when $\theta$ was 30°, there are two secondary vortices rotating around the central axis, and the area of the vortex core evolution to elliptical, which is a typical dual-celled vortex (Figure 7c). However, in Figure 8 ($r_0 =0$ cm), the vortex structure is transformed into a dual-celled vortex when *S* is only 0.18 and $\theta$ is 10°.

This indicates that the updraft radius has the more significant effect on the evolution of tornado vortex configuration from single vortex, to vortex breakdown, to vortex touchdown to multi-vortex.

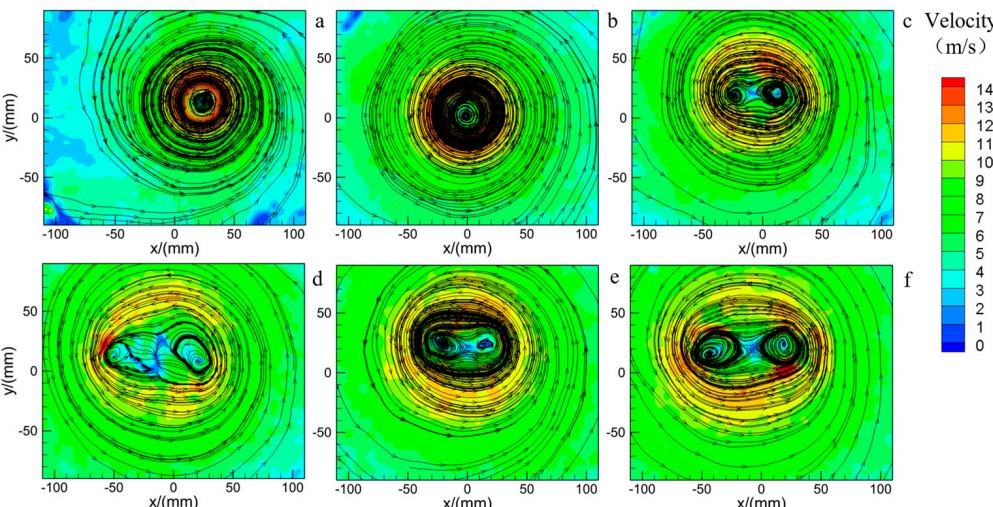

**Figure 7.** Instantaneous streamlines superimposed on the horizontal velocity contours of group 2: (**a**) $\theta = 10°$; (**b**) $\theta = 20°$; (**c**) $\theta = 30°$; (**d**) $\theta = 40°$; (**e**) $\theta = 50°$; (**f**) $\theta = 60°$.

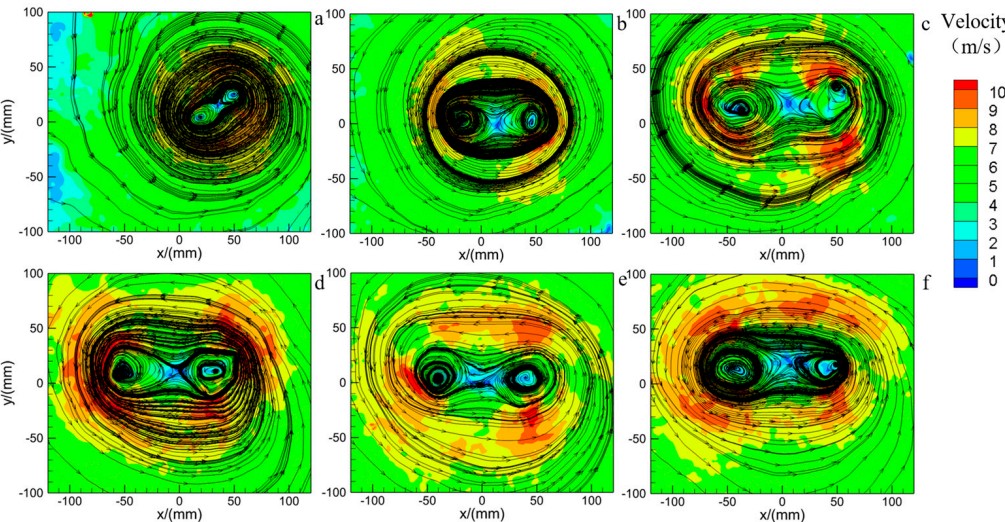

**Figure 8.** Instantaneous streamlines superimposed on the horizontal velocity contours of group 3: (**a**) $\theta = 10°$; (**b**) $\theta = 20°$; (**c**) $\theta = 30°$; (**d**) $\theta = 40°$; (**e**) $\theta = 50°$; (**f**) $\theta = 60°$.

## 5. Different Swirl Ratios Caused by the Updraft Radius

Further, the effect of the swirl ratio on the flow field is explored by varying the updraft radius at three fixed turning-vane angles. The grouping is shown in Table 3.

**Table 3.** Group sets with different updraft radius.

| Group | Angle of the Turning Vanes $\theta$ (°) | Updraft Radius $r_0$ (cm) | Swirl Ratio $S$ |
|---|---|---|---|
| 4 | 10 | 10, 12.5, 15, 17.5, 20 | 0.09, 0.11, 0.13, 0.15, 0.18 |
| 5 | 20 | 10, 12.5, 15, 17.5, 20 | 0.18, 0.23, 0.27, 0.32, 0.36 |
| 6 | 30 | 10, 12.5, 15, 17.5, 20 | 0.28, 0.36, 0.43, 0.51, 0.58 |

Figure 9 presents instantaneous horizontal velocity contours and streamlines under different swirl ratios of group 4. It can be clearly seen that as the swirl ratio increases, the vortex radius spreads and the velocity first increases and then decreases significantly, due to the vortex breakdown as discussed above. As the swirl ratio is 0.09~0.15, the vortex structure in the simulator is single-celled. While $S = 0.18$, three low-speed regions can be

observed in Figure 9e, corresponding to the vortex core observed in the streamlines, which demonstrates the single-celled vortex evolving into a dual-celled vortex.

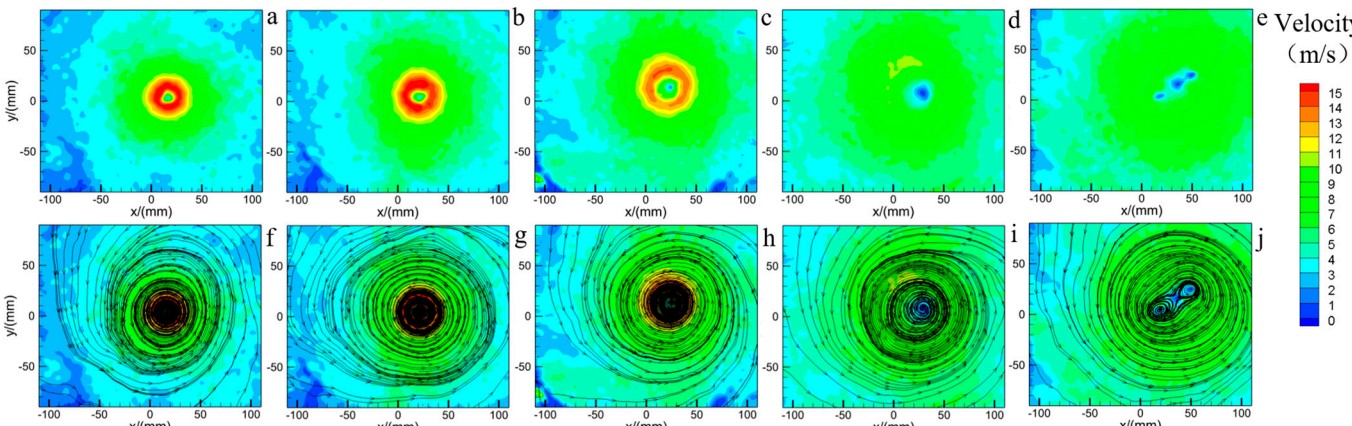

**Figure 9.** Instantaneous horizontal velocity contours and streamlines of group 4: (**a**) $r_0 = 0$ cm; (**b**) $r_0 = 12.5$ cm; (**c**) $r_0 = 15$ cm; (**d**) $r_0 = 17.5$ cm; (**e**) $r_0 = 20$ cm; (**f**) $r_0 = 10$ cm; (**g**) $r_0 = 12.5$ cm; (**h**) $r_0 = 15$ cm; (**i**) $r_0 = 17.5$ cm; (**j**) $r_0 = 20$ cm.

Figures 10 and 11 show the results of group 5 and group 6. With the increase of $S$, the three low-speed regions change into two more obvious low-speed regions, and one dominant vortex appears in the two secondary vortices, because it is consistent with the rotation direction of the peripheral air flow. More importantly, at $\theta = 20°$, when $r_0$ reaches 17.5 and $S$ reaches 0.32, the flow-field evolution to a dual-celled vortex. While $\theta = 30°$ cm, structural transformation occurs when $S$ is 0.43 and $r_0$ is 15.

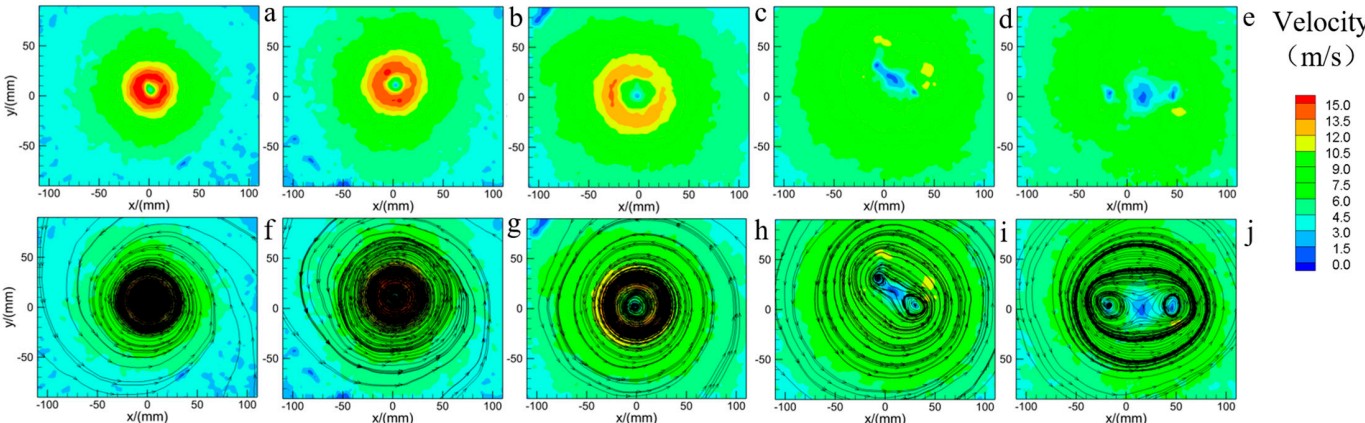

**Figure 10.** Instantaneous horizontal velocity contours and streamlines of group 5: (**a**) $r_0 = 10$ cm; (**b**) $r_0 = 12.5$ cm; (**c**) $r_0 = 15$ cm; (**d**) $r_0 = 17.5$ cm; (**e**) $r_0 = 20$ cm; (**f**) $r_0 = 10$ cm; (**g**) $r_0 = 12.5$ cm; (**h**) $r_0 = 15$ cm; (**i**) $r_0 = 17.5$ cm; (**j**) $r_0 = 20$ cm.

The above results show that the effects of the updraft radius and the angle of turning vanes on the tornado-like vortices are quite different, and the vortex structure changes such that the formation of the dual-celled vortex is more sensitive to the updraft radius, which results in the dual-celled vortex being more easily observed at larger updraft radius. For example, when $r_0 = 20$ cm, a dual-celled vortex can be observed at $\theta = 10°$ and $S = 0.18$. However, when $r_0 = 10$ cm, the flow field is always a single-celled vortex, although $S$ has reached 0.87 by adjusting the $\theta$ to 60°. The effect of increasing the angle of the turning vanes is to increase the angular momentum of the tornado-like vortices, thus enhancing the swirl ratio. However, increasing the updraft diameter enhances the angular momentum more significantly, creating a larger axial pressure gradient. Therefore, vortex breakdown and vortex transformation will occur at a lower swirl ratio.

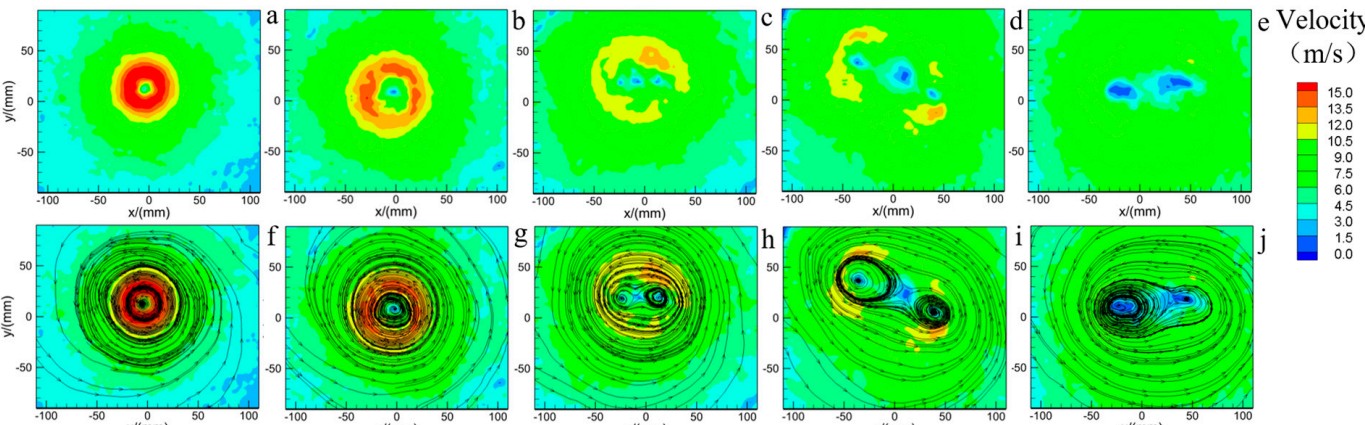

**Figure 11.** Instantaneous horizontal velocity contours and streamlines of group 6: (**a**) $r_0 = 10$ cm; (**b**) $r_0 = 12.5$ cm; (**c**) $r_0 = 15$ cm; (**d**) $r_0 = 17.5$ cm; (**e**) $r_0 = 20$ cm; (**f**) $r_0 = 10$ cm; (**g**) $r_0 = 12.5$ cm; (**h**) $r_0 = 15$ cm; (**i**) $r_0 = 17.5$ cm; (**j**) $r_0 = 20$ cm.

In addition, the phenomenon of vortex wandering in group 1 and group 4 is investigated. Wandering is defined as a random movement of the tornado-like vortex from its time-averaged position and, therefore, can influence vortex characteristics such as time-averaged velocities and core size. Here, the vortex-center locations in the horizontal planes when the flow field is single-celled are tracked as Figures 12 and 13, and 80 data are recorded in each image. The black dot is the instantaneous vortex center, and the red circle is the average vortex radius.

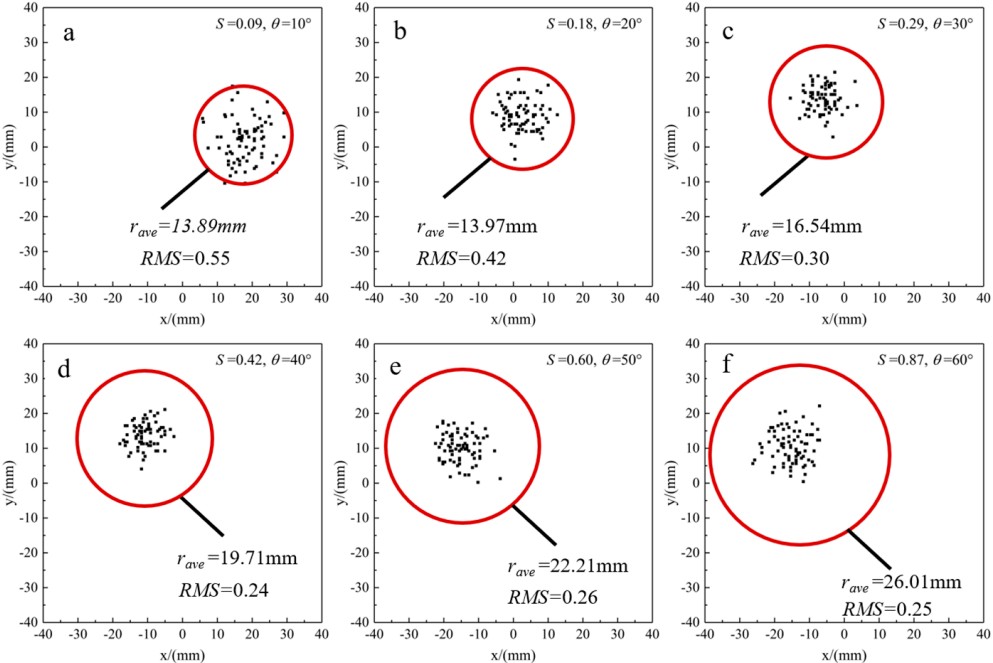

**Figure 12.** Instantaneous vortex core distribution and average vortex radius of group 1: (**a**) $\theta = 10°$; (**b**) $\theta = 20°$; (**c**) $\theta = 30°$; (**d**) $\theta = 40°$; (**e**) $\theta = 50°$; (**f**) $\theta = 60°$.

In both cases, at low *S*, the transient vortex centers are quite dispersed without any preference, with a small fraction of vortex centers located outside the vortex radius. And, the transient vortex centers are relatively concentrated at high *S*, none of the vortex center is out of the vortex radius [37]. Furthermore, the root mean square (RMS) of the radial distance between the instantaneous vortex center and the mean vortex center is employed to estimate the level of vortex wandering [38]. Lower RMS values indicate that vortex

wandering is suppressed. As observed, when changing $\theta$ from 10 to 60, the RMS value reduced from 0.55 to 0.25 by increasing $S$ from 0.09 to 0.87, and the suppression of vortex wandering becomes weaker when $S$ is larger. However, when changing $r_0$ from 10 to 17.5, the RMS value reduced from 0.55 to 0.29 by increasing $S$ from 0.09 to 0.15. That is, vortex stability is improved by increasing $S$, and the updraft radius has a greater influence on the vortex-wandering phenomenon compared to the angle of the turning vanes. The reason may be a tornado-like vortex with a higher updraft radius allows for the damping of flow fluctuations induced by turbulence associated with the surrounding shear layers, as hypothesized by Iungo et al. [39].

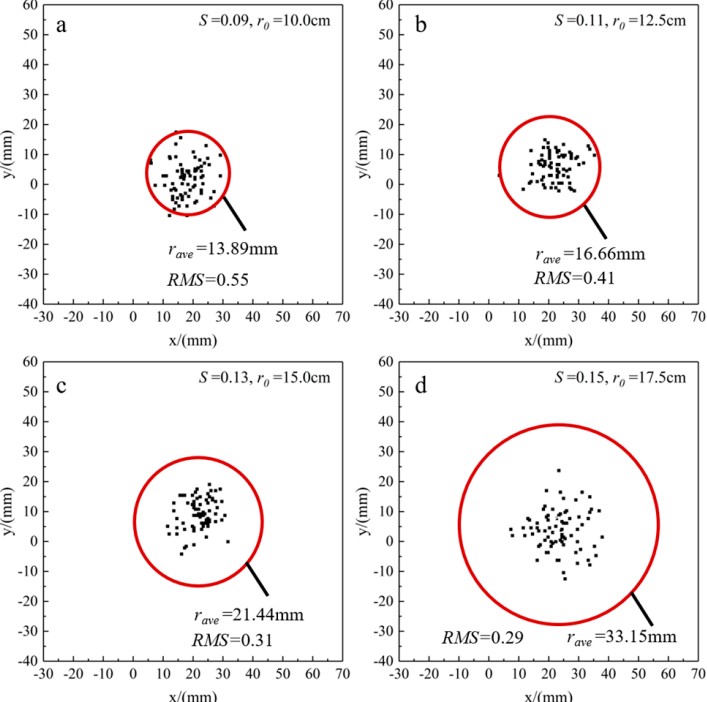

**Figure 13.** Instantaneous vortex core distribution and average vortex radius of group 4: (**a**) $r_0 = 10$ cm; (**b**) $r_0 = 12.5$ cm; (**c**) $r_0 = 15$ cm; (**d**) $r_0 = 17.5$ cm.

## 6. Conclusions

A new laboratory tornado simulator was constructed based on the traditional Ward-type tornado simulator, with a hexagonal prism structure that facilitates measurements using optical instruments such as PIV. The most important feature of the simulator is that the updraft radius can be changed independently. Hence, the effects of various swirl ratios caused by the updraft radius and the angle of entry flow on the tornado-like vortices can be investigated separately.

Firstly, the evolution of the tornado-like vortex was studied. When $r_0 = 20$ cm, a dual-celled vortex can be observed at $\theta = 10°$ and $S = 0.18$. However, when $r_0 = 10$ cm, the flow field was always single-celled vortex, although $S$ has reached 0.87 by adjusting the $\theta$ to 60°. Results showed that the formation of the dual-celled vortex is more sensitive to the updraft radius, since increasing the updraft diameter enhances the angular momentum more significantly, creating a larger axial pressure gradient. Simultaneously, Verma [18] demonstrated a gradual increase in the internal pressure of the tornado simulator as the outlet size was restricted and Zhang [40] observed the influence of inner cylinder radius on tornado shape transition and the formation of downdrafts using the ISU-type tornado simulator. These findings are consistent with our propositions in terms of pressure drop and velocity.

Additionally, vortex wandering was also analyzed. The results indicate that vortex stability is improved by increasing $S$, and the updraft radius has a greater influence on the

vortex-wandering phenomenon compared to the angle of the turning vanes. The results may be the flow fluctuations induced by turbulence.

Prospects for the future, considering that the existing simplified equation for swirl ratio may not be universally applicable to all Ward-type tornado simulators, particularly those with variable updraft radius, we propose the introduction of a novel mathematical factor into the equation to enhance the influence of the updraft radius parameter.

**Author Contributions:** Conceptualization, P.L.; validation, Y.W.; investigation, P.L., Y.Z. and Y.W.; data curation, Y.Z.; writing—original draft preparation, P.L.; writing—review and editing, B.W.; supervision, B.W. All authors have read and agreed to the published version of the manuscript.

**Funding:** This work was supported by China Postdoctoral Science Foundation (2022M711447), Gansu Province Youth Research Support Project (GXH202220530-13), Gansu Province Science and Technology Program Funding (22JR5RA519), Fundamental Research Funds for the Central Universities (lzujbky-2022-IT04).

**Institutional Review Board Statement:** Not applicable.

**Informed Consent Statement:** Not applicable.

**Data Availability Statement:** Data will be made available on request.

**Conflicts of Interest:** The authors declare no conflict of interest.

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
