# Peer review of "Experimental Investigation on the Influence of Swirl Ratio on Tornado-like Flow Fields by Varying Updraft Radius and Inflow Angle"

_atmosphere, doi:10.3390/atmos14091425_

Round 1
Reviewer 1 Report
An interesting paper, but much of the work reported is available elsewhere in the existing literature.
Reasonable.
Reviewer 2 Report
Review Comments to “Experimental Investigation on the Influence of Swirl Ratio on Tornado-Like Flow Fields by Varying Updraft Radius and Inflow Angle”, by Pengfei Lv , Yumeng Zhang * , Yanlei Wang , and Bo Wang *
In this manuscript, the authors reported the laboratory simulator experiments of the tornado-like flow structure changes with the changes of updraft radius and angle of entry flow. They concluded that, even with the same value of swirl ratio, when the combinations of updraft radius and angle of entry flow are different, there will be significant differences in the tornado-like flow structures. I believe the experiments are well designed and the findings are worth publication in the journal of Atmosphere. Therefore, I recommend this article to be published after some minor revisions. My specific comments are as follows.
1. The brief physical meaning of the radial Reynolds number, the swirl ratio and the aspect ratio should be earlier provided, better in the Introduction section. Then the mathematical expression can be provided in Section 2.
2. particle image velocimetry (PIV) is defined after its first appearance.
3. Line 90: “ Section 4” should be “Section 6”.
4. Line 163: “Fig. 11” should be “Fig.3”. Also in caption of Fig. 3 and the corresponding main text, please explain the differences between different panels for the same swirl ratios.
5. In Fig. 4, the curve of “Spencer and Mulhall tornados” are hard to see. The authors may try the “dot” type of curve instead.
6. In caption of Fig. 5, please clarify the corresponding turning vanes θ or Swirl ratio S in each subplot panel.
7. I am curious about the simplification of Swirl ratio (S) in Eq.(3). Is there any constrain on this derivation, for example, the updraft radius (r0) and angle of entry flow (θ) cannot be too small or too big? Otherwise, I feel that the physical meaning and the definition of the variable “Swirl ratio (S)” should be reconsidered. If so, the authors may direct raise this argument in the conclusion section.
Reviewer 3 Report
Minor Comments
Line 13: I recommend being more specific about what you mean “determining”. Specifically what about the vortex is determined from the swirl ratio? Also, it would not hurt to briefly define swirl ratio right away in the abstract just to refresh the reader, who may not be a specialist.
A reference conspicuously missing: https://www.sciencedirect.com/science/article/abs/pii/S0167610518305877
It may be worth comparing or remarking upon the results of this study in comparison with some of the recent LES and CFD experiments to explore tornado-like vortices. For example: https://www.sciencedirect.com/science/article/abs/pii/S016761052300171X
https://koreascience.kr/article/JAKO202134256006676.page
https://journals.sagepub.com/doi/abs/10.1177/13694332221119867 (results from your study compare with this one)
Lines 30-32: How much of this is due to simply better reporting now versus actual change?
Line 36: The literature in observations and modeling is vast! In modeling, there are mesoscale (e.g., Markowski’s, Finley’s, and other works related to supercell and non-supercell dynamics) and LES studies (e.g., Orf’s CM1 studies) and there are even climate change related studies that are related to tornado favorability parameters. I suppose it’s hard to do the background justice in the space constraints of the introduction. At least one major reference to Project VORTEX in various US regions should be mentioned.
Line 98: I think that you should mention this is the *Buckingham* pi theorem.
Figure 4: The observational lines are very difficult to see. Please enhance them and differentiate their color better.
Conclusions: What I feel is missing (and this is an optional suggestion) is a comparison and discussion of how the results here compare with observations and numerical simulation of tornadoes. Do we find the same swirl ratio and updraft size relationships in observed and LES simulations of tornadoes. Are the laboratory vortices and their experimental variations representing what would be found in nature?
Overall, there are a number of grammatical errors that need to be addressed before publication. I am unfortunately pressed for time to address them thoroughly in this review (due to unfortunate time constraints on my end, I focused primarily on the science), but I recommend having some proofreading done to improve the writing.
Reviewer 4 Report
Dear Authors,
Experimental Investigation on the Influence of Swirl Ratio on Tornado-Like Flow Fields by Varying Updraft Radius and Inflow Angle
This paper used a laboratory tornado simulator that was designed, constructed, and tested to generate tornado-like vortices. The authors found the effects of the updraft radius and the angle of turning vanes on the tornado-like vortices are quite different, and the formation of the dual-celled vortex is more sensitive to the updraft radius because a larger angular momentum and axial pressure gradient can be provided. The study deals with an important topic and aims to improve the understanding of tornadoes. I suggest a minor revision.
The paper is very clearly written.
The abstract needs to clarify the purpose of this paper.
The objectives are not well described.
Please check the journal's citation guidelines. And the presentation of references.
In general, the results are well presented and demonstrate the importance of the study.
Additionally, we encourage you to change the colour maps in your figures from jet to another map accessible to our colourblind readers.
Round 2
Reviewer 4 Report
Dear Authors,
The authors of the article entitled "Experimental Investigation on the Influence of Swirl Ratio on Tornado-Like Flow Fields by Varying Updraft Radius and Inflow Angle", meticulously incorporated all revisions suggested by all reviewers. His painstaking modifications significantly improved the study's clarity, coherence, and methodological rigor. Therefore, I am convinced of the robustness and relevance of the paper, and therefore recommend its publication. Their contributions provide a valuable perspective for understanding tornado-like flow patterns, enriching the research field and advancing significant advances in scientific knowledge.